# Association of Income with Post-Stroke Cognition and the Underlying Neuroanatomical Mechanism

**DOI:** 10.3390/brainsci13020363

**Published:** 2023-02-20

**Authors:** Jingyuan Tian, Yue Wang, Li Guo, Shiping Li

**Affiliations:** 1Department of Neurology, The Second Hospital of Hebei Medical University, Shijiazhuang 050051, China; 2Department of Neurology, Beijing Tiantan Hospital, Capital Medical University, Beijing 100069, China; 3National Clinical Research Center for Neurological Disease, Beijing Tiantan Hospital, Capital Medical University, Beijing 100069, China; 4Advanced Innovation Center for Human Brain Protection, Capital Medical University, Beijing 100069, China

**Keywords:** income, socioeconomic status, global brain atrophy, post-stroke cognition, post-stroke cognitive impairment

## Abstract

Objective: To investigate the association between income and post-stroke cognition at 3 months, and the underlying neuroanatomical mechanism. Methods: Patients with first-ever ischemic stroke were enrolled and analyzed. Baseline information on income and neuroimaging measurements with predictive values for post-stroke cognitive impairment (PSCI) were collected within 7 days of the admission. Three months after the index stroke, all participants underwent a detailed neuropsychological test battery. The associations between income and PSCI and between income and brain structural measurements were investigated. Results: A total of 294 patients were recruited for this study. Lower income was independently associated with poor cognitive performance on Stroop tests, Clinical Dementia Rating, Boston Naming Test, and Verbal Fluency Test. Regarding neuroimaging parameters, lower income was associated with a lower total brain volume (TBV)/total intracranial volume (TICV) ratio (*p* = 0.004). Conclusions: Lower income is associated with an increased chance of post-stroke cognitive decline, particularly in executive function and language domains. Since global brain atrophy (measured by TBV/TICV ratio) is a strong predictor for PSCI, its correlation with income may help explain the neuroanatomical mechanism between income and post-stroke cognition.

## 1. Introduction

Stroke is one of the leading causes of death and disability worldwide [1]. While the functional outcomes of ischemic stroke patients have improved with the rapid development of acute reperfusion therapy [2], post-stroke cognitive impairment (PSCI) remains prevalent, occurring in approximately half of survivors [3]. The negative impacts of PSCI on stroke recurrence [4], disability [5], and mortality [6] will impose substantial burdens on patients, their caregivers, and healthcare systems. Therefore, efforts should be made immediately to identify modifiable risk factors for PSCI for early disease detection and prompt intervention.

Cognitive function following a stroke varies significantly among individuals. The observed discrepancies may be caused by the interaction of multiple factors, including demographic characteristics, prior pathology, vascular risk factors, and stroke characteristics [7]. Among these factors, socioeconomic status (SES) is an active topic of research in cognitive decline and dementia [8,9], and its detrimental effects on poor functional outcomes, disability, and mortality in stroke survivors have been well investigated [10,11]. However, the literature exploring the association between SES and PSCI is scant, especially in developing countries such as China, where massive SES inequality exists. SES is a comprehensive concept with diverse components, and measures of individual-level SES typically consist of income, occupation, and educational attainment. A recent study suggested that these indicators are distinct theoretical constructs, likely with different pathways influencing and affecting cognitive functions [12]. Thus, disentangling the contributions of each indicator will provide important insights into the potential mechanisms of how SES shapes the risk of cognitive impairment and dementia. Greg J. Duncan et al. suggested utilizing the economic components of SES for monitoring relationships between SES and health [13]. Income is among the most important parts of the economic composition of SES, and studies have shown that lower income is a significant predictor of poorer cognitive function [14]. Individuals with annual per capita income below the poverty line are at a higher risk of developing cognitive impairment [15]. A longer duration of poverty, in both young adults and the elderly, was associated with poorer cognitive function [16,17]. As for stroke survivors, a recent study of the relationship between risk factors and several health domains revealed that lower income was associated with worse executive function scores [18]. Nevertheless, in this study, the Quality of Life in Neurological Disorders executive function scale, rather than a detailed neuropsychological assessment, was administered to patients. Thus, the effect of income on domain-specific cognitive functions after stroke remains to be elucidated.

A growing body of literature has shown that SES indicators are associated with specific brain structures. For example, community socioeconomic disadvantage was associated with cortical morphology [19]. Educational attainment moderates the effect of age on hippocampal volume [20]. Lower socioprofessional attainment at midlife was associated with faster hippocampal atrophy [21]. Events associated with lower socioeconomic environments during development and aging could influence the structural and functional plasticity of the hippocampus, amygdala, and prefrontal cortex [22]. Interestingly, some of these brain structures are also known to have the potential to predict PSCI in the acute phase of the stroke. Understanding the association between income and magnetic resonance imaging (MRI) predictors for PSCI will help understand the potential neuroanatomical mechanisms of income-related post-stroke cognitive performance. According to a recent systematic review, among MRI parameters with predictive values for cognitive outcomes after stroke, global brain atrophy is one of the most consistent predictors across various studies [23]. Brain atrophy refers to the loss or shrinkage of neurons, which is reflected as a reduction in brain volume, and can be measured using neuroimaging. In clinical practice, the ratio of total brain volume (TBV) to total intracranial volume (TICV), also known as the brain parenchymal fraction, is frequently used to measure global brain atrophy [24]. TBV/TICV ratio was found to be associated with neuropsychological performance [25] and is a simple technique to distinguish dementia patients from healthy controls [26]. Although a positive correlation between SES and brain volume in aging-related regions has been identified [27], the relationship between income and total brain volume or global brain atrophy (as indexed by TBV/TICV ratio) in stroke survivors is yet unknown.

Based on the evidence of the relationship between global brain atrophy and PSCI, if we can further demonstrate a connection between income and PSCI, as well as between income and global brain atrophy, we can infer that income may be correlated with PSCI via the brain atrophy process. To test this hypothesis, we utilized data from the China National Clinical Research Center Alzheimer’s Disease and Neurodegenerative Disorder Research (CANDOR), which included information on first-ever ischemic stroke patients. Baseline information about income and MRI parameters were collected within 7 days post-stroke. Three months after the index stroke, all the participants underwent a battery of neuropsychological tests. After categorizing participants into lower- and higher-income groups, neuropsychological features and MRI measurements were compared, and the effects of income on each cognitive domain and structural brain measurement were investigated.

## 2. Materials and Methods

This study was a subgroup analysis of CANDOR. CANDOR is an ongoing study started in July 2019, with the goal being to investigate the underlying mechanisms, risk factors, and prognosis of cognitive impairment and dementia. Individuals eligible for this study included the following criteria: (1) aged between 40 and 100, (2) hospitalized with first-ever ischemic stroke, (3) duration from the stroke onset to admission being less than 7 days, and (4) without pre-existing cognitive impairment, indicated by a score of ≤3.5 from the Informant Questionnaire for Cognitive Decline. Exclusion criteria were as follows: (1) a history of asymptomatic cerebral infarction, psychiatric disorders, uncontrolled epilepsy, severe pulmonary and cardiovascular diseases, systematic diseases that affect the central nervous system, uncontrolled metabolic syndrome and endocrine diseases, and malignant tumors, and (2) poor ability to fulfill the examinations, such as literacy problems, severe hearing difficulty, and visual impairment. Data of the subjects were initially obtained from the vascular cognitive cohort of the CANDOR study, based on the above eligibility criteria. After excluding individuals who did not provide information on their monthly income, and those with missing 3D-T1 MRI data and incomplete neuropsychological assessment, 294 patients were finally enrolled in this study.

We used monthly income as a proxy for SES. To avoid an illness factor on income, we recorded the data of self-reported pre-stroke income during their hospitalization. Monthly income was calculated by summing up all the households’ monthly income and dividing it by the number of residents. Participants were separated into the following two groups: (1) those with a monthly income of less than 5000 Chinese Yuan (CNY), and (2) those with a monthly income of more than CNY 5000. The group classification was not arbitrary. We used the per capita disposable income data in Beijing for reference, which was CNY 69,433.5 in 2020. We then averaged the per capita disposable income every month (CNY ≈ 5786) and used CNY 5000 for the group criteria. The information on the population’s income is available from the China Statistical Yearbook, published by the National Bureau of Statistics of China. Other variables assessed at baseline contained demographic characteristics and vascular risk factors. Demographic characteristics were age, gender, and years of educational attainment. Vascular risk factors included history of hyperlipidemia, diabetes, hypertension, smoking, and drinking.

Each research center quantitatively measured signal-to-noise ratio, uniformity, and geometric distortion. MRI data were acquired using a 3.0 T scanner, with the maximum thickness of 1.5 mm. The 3D T1-weighted anatomical images were corrected for intensity and non-uniformity with the N4 algorithm. Reconstruction of the brain surface was acquired via the FreeSurfer (version 7.2.0, http://surfer.nmr.mgh.harvard.edu/) (accessed on 6 October 2022) recon-all pipeline. The following brain measurements were all obtained with this pipeline: hippocampus volume, TBV, cortex volume, white matter volume, gray matter volume, TBV, TICV, TBV/TICV ratio, hippocampus volume, and mean cortical thickness. TBV was calculated by summing the volume of gray matter and white matter. TICV was calculated by adding the TBV and the volume of cerebrospinal fluid. TBV/TICV ratio was computed as the percentage of TBV to TICV to correct for differences in head size. Global brain atrophy was represented by a lower TBV/TICV ratio.

All the participants underwent a battery of neuropsychological tests 3 months after a stroke, which included the following aspects: (1) tests for global cognitive performance, such as Mini-Mental State Examination (MMSE), Montreal Cognitive Assessment (MoCA), and Clinical Dementia Rating (CDR); (2) tests for domain-specific cognitive performance, including Digit-Span test (DST), Rey Auditory Verbal Learning Test (RAVLT), Rey-Osterrieth Complex Figure Test (ROCF), Stroop Color-Word Test-Victoria version, Trail Making Test part A and B (TMT A and TMT B), Symbol Digit Modalities Test (SDMT), Verbal Fluency Test-animal (VFT), Boston Naming Test (BNT), and Clock Drawing Test (CDT); and (3) tests for psychiatric symptoms, including Neuropsychiatric Inventory (NPI) and Geriatric Depression Scale (GDS). As for domain-specific cognitive tests, words in DST backwards and DST forwards are summed to obtain DST total score. RAVLT in our study includes 3 sub-tests, RAVLT total learning, RAVLT long-delayed recall, and RAVLT recognition. ROCF has 4 sub-tests, including ROCF copy, ROCF immediate recall, ROCF long-delayed recall, and ROCF recognition [28]. The Victoria version Stroop tests consists of 3 tests, including Dot (D) test, Word (W) test, and Color (C) test. Time to complete each test was scored [29]. The obtained score of TMT A and TMT B represent the amount of time required to complete the task [30]. The score of SDMT was the number of correctly matched digits and symbols in a given time (90 s) [31]. VFT was determined by the number of animal names in 1 min. BNT Beijing version included 30 pictures, and the number of correctly named pictures were scored. CDT required subjects to draw a clock pointing to 11:10 on a white sheet of paper [32]. Lower values indicate better performance in CDR, Stroop test, TMT A and B, NPI, and GDS, but poor performance in MMSE, MoCA, DST, RAVLT, ROCF, SDMT, VFT, BNT and CDT. All the tests were administered by examiners trained in neuropsychological assessment.

Descriptive analyses were performed to summarize the characteristics of the participants. Continuous variables were expressed as means (standard deviations) and categorical variables as counts (percentages). Potential differences between two income groups were tested by 2-tailed t tests for continuous data and chi-squared tests for categorical data. To summarize the relationship between income and variables of cognitive performance and MRI outcomes that showed significant differences in group comparisons, multiple linear regression models were performed to calculate the β coefficient and accompanying 95% CIs. All the models were adjusted for age, gender, and years of educational attainment. We used SPSS software (version 24.0) for the analyses. A 2-sided *p* value <0.05 was considered statistically significant.

## 3. Results

Two hundred ninety-four first-ever ischemic stroke patients were evaluated in this study, of whom 178 patients had a monthly income of less than CNY 5000, and 116 had a monthly income of more than CNY 5000. The mean age of the whole sample was 58.3 years (±9.2), 226/294 (76.9%) were male, and mean years of educational attainment was 10.7 (±3.3). Full characteristics of the study participants are described in Table 1.

### 3.1. Comparisons between Patients with Lower Income and Higher Income

Table 1 shows the differences in demographics and vascular risk factors between the two groups. Patients in the lower-income and higher-income groups did not differ in age, gender, or vascular risk factors. Compared to higher-income group patients, lower-income patients had fewer years of schooling (*p* < 0.001). Comparisons of neuropsychological assessments 3 months after the stroke are presented in Table 2. Patients in the lower-income group showed lower scores on MMSE, MoCA, DST total, RAVLT total learning, VFT, and BNT, and higher scores on CDR and Stroop tests (all *p* < 0.05).

MRI outcomes in each income group are displayed in Table 3. Patients in the lower-income group had a lower TBV/TICV ratio (*p* = 0.022) compared to their higher-income counterparts. However, the two groups had no significant differences in bilateral cortex volume, white matter volume, gray matter volume, TBV, bilateral hippocampus volume, and bilateral mean cortical thickness.

### 3.2. Association between Income and Post-Stroke Cognitive Functions and MRI Outcomes

Linear regression models were used to assess the association of income with post-stroke cognitive performance and MRI outcomes, which showed statistical differences in group comparisons. As shown in Table 4, after adjustments for age, gender, and educational years, as the level of income increased, the risks of lower scores on global CDR (β = −0.120, 95%CI: −0.163 to −0.005), total CDR (β = −0.147, 95%CI: −0.689 to −0.088), Stroop D time (β = −0.163, 95%CI: −5.837 to −0.951), Stroop W time (β = −0.158, 95%CI: −11.153 to −1.517), and Stroop C time (β = −0.144, 95%CI: −9.181 to −0.946) increased, and the risks of higher scores on VFT (β = 0.142, 95%CI: 0.308 to 2.630) and BNT (β = 0.113, 95%CI: 0.060 to 1.811) also increased. The only neuroimaging parameter in the multivariable linear regression model was the TBV/TICV ratio. A positive association was found between income and TBV/TICV ratio (β = 0.166, 95%CI: 0.004 to 0.024), when controlled for age, gender, and years of education.

## 4. Discussion

In this study, we found that income was independently associated with cognitive performance 3 months after the stroke and the TBV/TICV ratio (as a proxy for global brain atrophy). Given that global brain atrophy has been demonstrated to have a predictive value for PSCI in previous studies [23,33], our findings may help reveal the neuroanatomical underpinnings of the association between income and post-stroke cognitive performance.

The association between lower income and worse cognitive performance 3 months after the ictus found in our study was consistent with a prior review, which showed that stroke survivors from lower SES had worse outcomes [34]. Regarding post-stroke cognitive outcomes, Abdel Douiri et al. found PSCI prevalence increased as the level of SES decreased [35]. In contrast to that study, which utilized employment as a proxy for SES, our study provided new evidence for the detrimental effect of socioeconomic disadvantage on post-stroke cognitive performance, by using income as an indicator of SES. Furthermore, we performed detailed neuropsychological tests, which enabled us to identify income-related post-stroke cognitive decline in CDR, Stoop tests, VFT, and BNT. CDR is a widely used cognition severity-ranking scale [36], and the remaining tests focus mainly on executive function and language ability [37,38]. Compared to the previous study, which indicated a close relationship between income and executive deficits in stroke survivors [18], we further discovered that the Stroop tests were sensitive in assessing lower income-related post-stroke executive deficits, and that lower income also had a negative influence on both global cognition severity and language ability. Executive dysfunction is a recognized feature of vascular cognitive impairment [39], and impairment in executive function could predict poor survival in ischemic stroke patients [6]. Thus, our findings of lower income as a modifiable risk factor for executive function in stroke survivors have profound implications.

Due to the strong association between specific neuroimaging markers and PSCI, we next investigated the effects of lower income on several brain structures with previously demonstrated predictive values in PSCI. Here, we showed that income was significantly associated with global brain atrophy. This result is consistent with those studies indicating that SES influences brain integrity. For example, Leslie Grasset et al. revealed that income volatility, which implied drops in income and episodes of lower income, was associated with total brain microstructural integrity [40]. Moreover, in a cross-sectional study, Hunt et al. found that a high level of neighborhood socioeconomic disadvantage was associated with considerably decreased TBV [41]. Another study demonstrated that higher SES was associated with greater global brain volumes in white adults [42]. There could be several explanations for the association between income and global brain atrophy. First, exposure to low income can be regarded as a stressful event that may impact the brain’s plasticity via neurobiological pathways [22]. For instance, stress-related disruption of hypothalamic–pituitary–adrenal (HPA) axis activity may affect brain areas with corticosteroid receptors. Indeed, this type of receptor is present throughout the whole brain. Prior research has shown a correlation between higher cortisol (output of the HPA axis) levels and decreased total brain-tissue volume [43]. Second, lifelong exposure to low-income conditions may lead to unhealthy behaviors. It has been demonstrated that smoking for an extended period of time severely affects brain perfusion levels [44], which may result in reduced cerebral volume. High-income individuals, in contrast, typically have healthy lifestyles, which includes getting more exercise. Physical activity has been linked to less brain atrophy [45] and may improve brain health by increasing neurotrophic factor production and brain plasticity [46].

After demonstrating the relationship between income and post-stroke cognitive decline and between income and TBV/TICV ratio, we tried to develop a framework that connected income, global brain atrophy, and post-stroke cognitive performance by interpreting the association between global brain atrophy and PSCI. Previously, a rich body of literature described the association between brain atrophy and PSCI. Majon Muller et al. indicated that smaller TBV was associated with poor executive performance, and this association strengthened with infarcts [47]. Another study found a cumulative effect of global brain atrophy on cognition in dementia-free elderly with cerebrovascular disease [48]. Furthermore, a Singapore study using moderated mediation analysis demonstrated that global atrophy was indirectly related to post-stroke dementia by disrupting language, executive functions, and memory [49]. The underlying mechanism between brain atrophy and PSCI can be explained as follows. First, global brain atrophy reflects a disease- or aging-associated loss of the brain. According to the brain reserve concept, a pre-existing neuropathology may impact the brain’s capacity to utilize the remaining brain tissue to take over functions from areas affected by the subsequent neuropathology (e.g., stroke attack) on cognition [50]. Second, concerning executive function, previous studies have provided insights into how income and other SES indicators affect executive functions through brain structural changes. Executive functions are mainly subserved by the prefrontal cortex. As mentioned earlier, stressful life events caused by lower income could alter the plasticity of the whole brain. The prefrontal cortex is one of the most vulnerable brain structures during this process [22]. This conclusion was further validated by a recent study showed that a SES-related decline in executive function was mediated by a reduction in dorsolateral prefrontal cortex volume [51]. According to the “bigger is better” hypothesis, loss of prefrontal cortex volume was associated with worse executive function [52]. Meanwhile, patients with a smaller prefrontal cortex cannot resist subsequent stroke attacks on executive functions.

The strength of this study Is the”comb’nation of structural MRI measurements and comprehensive neuropsychological tests on a group of first-ever ischemic stroke patients. As far as we know, we are the first study to explore the neuroanatomical correlates of income and post-stroke cognitive performance. Several limitations should be noted in this study. First, given that the income and neuroimaging data were collected at the baseline, we cannot demonstrate the causality of the observed relationship between income and global brain atrophy. Future prospective studies with follow-up data are needed to determine the causality between them and how the brain structure will evolve during exposure to decreased income. Second, because the follow-up of CANDOR is ongoing, we only obtained the data on cognitive performance 3 months after the stroke. It is necessary to further explore the effect of income on long-term post-stroke cognitive performance. Finally, income was recorded as self-based data, which might contain recall bias. However, self-reported income is commonly utilized in research that evaluates the association between income and health.

## 5. Conclusions

Lower income is associated with worse post-stroke cognitive performance, including global cognitive function, executive function, and language ability. Exposure to lower income may lead to cognitive decline in stroke survivors by disrupting pre-stroke normal brain aging. Our findings have important implications. Along with secondary prevention for stroke recurrence, public health promotion strategies for PSCI prevention should be modified to narrow the income inequalities and protect those particularly disadvantaged.

## Figures and Tables

**Table 1 brainsci-13-00363-t001:** Characteristics of Study Participants by Income.

Variables	Total (*n* = 294)	Income ≤ 5000 (*n* = 178)	Income > 5000 (*n* = 116)	*p* Value
Age, mean (SD), y	58.3 (9.2)	57.8 (9.1)	59.0 (9.4)	0.269
Gender, male, *n* (%)	226 (76.9%)	134 (75.3%)	92 (79.3%)	0.423
Education, mean (SD), y	10.7 (3.3)	9.9 (2.9)	12.0 (3.5)	<0.001
Diabetes, *n* (%)	93 (31.6%)	51 (28.7%)	42 (36.2%)	0.173
Hypertension, *n* (%)	167 (56.8%)	98 (55.1%)	69 (59.5%)	0.454
Hyperlipidemia, *n* (%)	64 (21.8%)	40 (22.5%)	24 (20.7%)	0.717
Smoking, *n* (%)	132 (44.9%)	80 (44.9%)	52 (44.8%)	0.984
Drinking, *n* (%)	107 (36.4%)	63 (35.4%)	44 (37.9%)	0.658

**Table 2 brainsci-13-00363-t002:** Comparisons of Neuropsychological Assessments.

Neuropsychological Tests ^a^	Income ≤ 5000	Income > 5000	*p* Value
MMSE	23.9 (4.6)	25.5 (2.8)	<0.001
MoCA	19.8 (5.3)	21.6 (4.5)	0.003
Global CDR score	0.3 (0.4)	0.2 (0.2)	<0.001
Total CDR score	1.0 (1.5)	0.5 (0.8)	<0.001
DST total	10.8 (2.8)	11.7 (2.8)	0.004
RAVLT total learning	32.1 (11.4)	35.8 (12.0)	0.008
RAVLT long-delayed recall	5.4 (3.7)	6.0 (4.0)	0.149
RAVLT recognition	7.7 (7.3)	7.7 (8.5)	0.928
ROCF copy	28.6 (32.0)	24.6 (11.9)	0.395
ROCF immediate recall	13.0 (10.3)	11.7 (10.4)	0.475
ROCF long-delayed recall	11.9 (10.0)	11.1 (10.3)	0.669
ROCF recognition	17.7 (3.1)	17.9 (3.3)	0.441
Stroop D time	25.7 (11.4)	21.5 (7.7)	<0.001
Stroop W time	34.9 (23.6)	27.9 (10.2)	0.001
Stroop C time	41.0 (19.7)	35.0 (12.2)	0.002
TMT A	62.5 (31.0)	54.8 (34.4)	0.173
TMT B	139.5 (98.3)	125.0 (82.2)	0.363
SDMT	27.5 (13.6)	31.5 (14.6)	0.082
VFT	14.1 (4.8)	16.4 (5.2)	<0.001
BNT	21.0 (4.2)	22.8 (3.5)	<0.001
CDT	8.0 (2.5)	8.3 (2.1)	0.210
NPI	2.0 (5.9)	1.3 (2.9)	0.261
GDS	3.0 (2.7)	3.1 (2.7)	0.808

^a^—results are expressed as mean (SD).

**Table 3 brainsci-13-00363-t003:** Comparisons of MRI outcomes.

MRI Outcomes	Income ≤ 5000	Income > 5000	*p* Value
Left cortex volume ^a^	224,321.8 (21,002.0)	221,565.4 (20,003.6)	0.263
Right cortex volume ^a^	223,867.0 (21,842.3)	221,201.2 (19,530.9)	0.287
White matter volume ^a^	457,446.2 (54,594.2)	453,105.5 (51,407.8)	0.496
Gray matter volume ^a^	604,332.8 (53,542.4)	597,600.6 (50,852.4)	0.283
TBV ^a^	1,117,680.1 (105,441.6)	1,108,126.8 (101,394.5)	0.441
TBV/TICV ratio ^b^	74.4 (3.8)	75.5 (4.6)	0.022
Left hippocampus volume ^a^	3523.2 (363.1)	3483.8 (408.4)	0.387
Right hippocampus volume ^a^	3652.4 (401.9)	3573.4 (370.8)	0.091
Left mean cortical thickness ^c^	2.4 (0.1)	2.4 (0.1)	0.352
Right mean cortical thickness ^c^	2.4 (0.1)	2.4 (0.1)	0.399

^a^—results are expressed as mean (SD), mm^3^; ^b^—results are expressed as mean (SD), %; ^c^—results are expressed as mean (SD), mm.

**Table 4 brainsci-13-00363-t004:** The linear regression models for the relationship between income and neuropsychological tests and MRI outcomes.

Dependent Variables ^a^	Standardized β Coefficient	95%CI	*p* Value
MMSE	0.094	(−1.108 to 1.653)	0.085
MoCA	0.061	(−0.413 to 1.663)	0.237
Global CDR score	−0.120	(−0.163 to −0.005)	0.038
Total CDR score	−0.147	(−0.689 to −0.088)	0.011
DST total	0.110	(−0.026 to 1.277)	0.060
RAVLT total learning	0.086	(−0.490 to 4.604)	0.113
Stroop D time	−0.163	(−5.837 to −0.951)	0.007
Stroop W time	−0.158	(−11.153 to −1.517)	0.010
Stroop C time	−0.144	(−9.181 to −0.946)	0.016
VFT	0.142	(0.308 to 2.630)	0.013
BNT	0.113	(0.060 to 1.811)	0.036
TBV/TICV ratio	0.166	(0.004 to 0.024)	0.004

^a^—all the models are adjusted by age, gender, and years of educational attainment.

## Data Availability

The datasets generated and/or analyzed during the current study are available from the corresponding author upon reasonable request.

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
