# Peer review of "Association of Income with Post-Stroke Cognition and the Underlying Neuroanatomical Mechanism"

_brainsci, 2023, doi:10.3390/brainsci13020363_

Round 1

Reviewer 1 Report

Comments and Suggestions for Authors

Thanks for recommending me as a reviewer. Thank you for recommending me as a reviewer. In this paper, the authors investigated the association between income and cognition of stroke at 3 months and the underlying neuroanatomical mechanisms. If the authors complete minor revisions, the quality of the study will improve.

1. The introduction section is well written. If the authors describe the theoretical background related to the effect of income on post-stroke cognition and the underlying neuroanatomical mechanism in more detail in the introductory section, it will help readers understand.

2. line 88: "2. Materials and Methods" - Authors should describe in more detail the selection criteria and sampling of subjects in the Methods section.

3. Authors should be more specific about their definition of SES in the Methods section.

Author Response

Dear reviewer,

Thank you very much for your comments and professional advice. These opinions help to improve academic rigor of our article. Based on your suggestion and request, we have made corrected modifications on the revised manuscript. Furthermore, we would like to show the details as follows:

Point 1: The introduction section is well written. If the authors describe the theoretical background related to the effect of income on post-stroke cognition and the underlying neuroanatomical mechanism in more detail in the introductory section, it will help readers understand.

Response 1: We think this is an excellent suggestion. Yes, it would be more understandable if we describe the theiretical background in detail. To the best of our knowledge, only one recent article has explored the influence of income on cognitive function in stroke survivors. Meanwhile, as reviewed by a study in 2015, socioeconomic status indicators used in exploring the relationship with cognitive impairments in stroke survivors only including the index of multiple deprivation, occupational backgrounds, and education. This is why we were interested in the correlation of income and post-stroke cognitive function. Thus, we could only add some recent literatures on the effect of income on cognitive impairment not specific to stroke survivors to help readers understand the background. Please see the added references:

① Miu, J.; Negin, J.; Salinas-Rodriguez, A.; Manrique-Espinoza, B.; Sosa-Ortiz, A.L.; Cumming, R.; Kowal, P. Factors Associated with Cognitive Function in Older Adults in Mexico. Global Health Action 2016, 9, 30747, doi:10.3402/gha.v9.30747.

② Chen, L.; Cao, Q. Poverty Increases the Risk of Incident Cognitive Impairment among Older Adults: A Longitudinal Study in China. Aging & Mental Health 2020, 24, 1822–1827, doi:10.1080/13607863.2019.1663491.

③ Zeki Al Hazzouri, A.; Elfassy, T.; Sidney, S.; Jacobs, D.; Pérez Stable, E.J.; Yaffe, K. Sustained Economic Hardship and Cognitive Function: The Coronary Artery Risk Development in Young Adults Study. American Journal of Preventive Medicine 2017, 52, 1–9, doi:10.1016/j.amepre.2016.08.009.

④ Yu, X.; Zhang, W.; Kobayashi, L.C. Duration of Poverty and Subsequent Cognitive Function and Decline Among Older Adults in China, 2005–2018. Neurology 2021, 97, e739–e746, doi:10.1212/WNL.0000000000012343.

Although lots of studies have indicated that economic stress caused by lower income could affect brain structures through hypothalamic-pituitary-adrenal axis, there are no separate articles exploring the effects of income on brain structure before. We therefore added some articles on the effects of other socioeconomic status indicators on brain structures to help readers to understand. Please see the added references:

① Gianaros, P.J.; Kuan, D.C.-H.; Marsland, A.L.; Sheu, L.K.; Hackman, D.A.; Miller, K.G.; Manuck, S.B. Community Socioeconomic Disadvantage in Midlife Relates to Cortical Morphology via Neuroendocrine and Cardiometabolic Pathways. Cereb. Cortex 2015, bhv233, doi:10.1093/cercor/bhv233.

② Noble, K.G.; Grieve, S.M.; Korgaonkar, M.S.; Engelhardt, L.E.; Griffith, E.Y.; Williams, L.M.; Brickman, A.M. Hippocampal Volume Varies with Educational Attainment across the Life-Span. Front. Hum. Neurosci. 2012, 6, doi:10.3389/fnhum.2012.00307.

③ Elbejjani, M.; Fuhrer, R.; Abrahamowicz, M.; Mazoyer, B.; Crivello, F.; Tzourio, C.; Dufouil, C. Life-Course Socioeconomic Position and Hippocampal Atrophy in a Prospective Cohort of Older Adults. Psychosom Med 2017, 79, 14–23, doi:10.1097/PSY.0000000000000365.

④ McEwen, B.S.; Gianaros, P.J. Central Role of the Brain in Stress and Adaptation: Links to Socioeconomic Status, Health, and Disease: Central Links between Stress and SES. Annals of the New York Academy of Sciences 2010, 1186, 190–222, doi:10.1111/j.1749-6632.2009.05331.x.

Point 2: line 88: "2. Materials and Methods" - Authors should describe in more detail the selection criteria and sampling of subjects in the Methods section.

Response 2: Thank you for your valuable suggestions. we added the following details:

“Data of subjects were initially obtained from the vascular cognitive cohort of the CANDOR study based on the above eligibility criteria. After excluding individuals without the information of monthly income, and those with missing 3D-T1 MRI data and incomplete neuropsychological assessment, 294 patients were finally enrolled in this study.”

Point 3: Authors should be more specific about their definition of SES in the Methods section

Response 3: Thank you for pointing this out. We have revised the definition of SES in the Methods section. The details are as follows:

”We used monthly income as a proxy for SES. To avoid an illness factor in income, we recorded the data of self-reported pre-stroke income during their hospitalization. Monthly income was calculated by summing up all the household’s monthly income and dividing it by the number of residents.”

Thank you very much for your attention and time. Look forward to hearing from you.

Yours sincerely,

1st Feb. 2023

Reviewer 2 Report

Comments and Suggestions for Authors

The topic is interesting, but I have some comments.

1.         Abstract It seems that cognitive performance of patients before stroke is uncertain in this study. In this case, how did you define cognitive decline? It is not certain whether cognitive function declined or not for a patient in this study. In addition, it is considered that cognitive function differs depending on income before stroke.

2.         AbstractWhat kind of statistical analysis was conducted in this study?

3.         Materials and MethodsIt is written hat baseline information about MRI parameters were collected within 7 days post-stroke in line 82. Could you explain more about it in the Methods section?

4.         Materials and Methods, line 145Why didn’t the authors adjust the other characteristics in the regression analysis?

5.         Materials and MethodsThere are so many outcomes. Could you add more information for the outcomes?

6.         (Results) How many patients were excluded from participants of CANDOR?

7.         (Results, line 149) “Nighty-four”

8.         (Discussion) As the authors wrote in the limitation, causal association is uncertain in this study. Therefore, “effect” in the title should be replaced with “association”.

Author Response

Dear reviewer,

Thank you very much for your comments and professional advice. These opinions help to improve academic rigor of our article. Based on your suggestion and request, we have made corrected modifications on the revised manuscript. Furthermore, we would like to show the details as follows:

Point 1: (Abstract) It seems that cognitive performance of patients before stroke is uncertain in this study. In this case, how did you define cognitive decline? It is not certain whether cognitive function declined or not for a patient in this study. In addition, it is considered that cognitive function differs depending on income before stroke.

Response 1: Thanks for your suggestion. Although we used the Informant Questionnaire on Cognitive Decline in the Elderly as a screening tool to exclude patients with pre-stroke cognitive impairment at the time of enrollment, we did not acquire their domain-specific cognitive functions before the ictus. We changed some of the use of the term “cognitive decline” in our article. In the Abstract section, multiple linear regression analysis showed that income was positively correlated with cognitive performance, thus we speculate that with the level of income decreased, the chance of post-stroke cognitive decline will be increased.

Point 2:(Abstract)What kind of statistical analysis was conducted in this study?

Response 2: Descriptive statistical analyses were first performed for all enrolled patients and for patients grouped according to income criteria (those with a monthly income of less than 5,000 Chinese Yuan and those with a monthly income of more than 5,000 Chinese Yuan). The cognitive tests and brain structure measurements with statistical differences in group comparisons were selected as dependent variables to analyze the effect of income on them (multiple linear regression analysis).

Point 3: (Materials and Methods)It is written hat baseline information about MRI parameters were collected within 7 days post-stroke in line 82. Could you explain more about it in the Methods section?

Response 3: Your suggestion really means a lot to us and readers. However, CANDOR is an ongoing study, the protocols of neuroimaging methods have not been fully published so far. Therefore, in the Methods section, we could only introduce the information of MRI data processing and analysis, and not mention the method of MRI data acquisition, which is similar to our previously published study of CANDOR :

Wang, Y.; Wang, S.; Zhu, W.; Liang, N.; Zhang, C.; Pei, Y.; Wang, Q.; Li, S.; Shi, J. Reading Activities Compensate for Low Education-Related Cognitive Deficits. Alzheimers Res Ther 2022, 14, 156, doi:10.1186/s13195-022-01098-1.

Point 4: (Materials and Methods, line 145)Why didn’t the authors adjust the other characteristics in the regression analysis?

Response 4: We think this is an excellent question. Age, gender, and education were the most common confounding factors when studying cognitive functions. As for post-stroke cognition, some previous studies further adjusted for variables about stroke characteristics, such as stroke severity, location, and vascular risk factors. In our study, vascular risk factors did not show significant differences between 2-level income group comparisons. In addition, to complete a battery of cognitive assessments which require some extent of comprehension, visual, hearing, and language ability, most of our participants are minor stroke survivors. Previously, studies have indicated that cognitive impairment across multiple domains is common following minor stroke regardless of infarct location:

Marsh, E.B.; Khan, S.; Llinas, R.H.; Walker, K.A.; Brandt, J. Multidomain Cognitive Dysfunction after Minor Stroke Suggests Generalized Disruption of Cognitive Networks. Brain and Behavior 2022, 12, doi:10.1002/brb3.2571.

Therefore, we did not adjust further for stroke-related characteristics in this study.

Point 5: (Materials and Methods)There are so many outcomes. Could you add more information for the outcomes?

Response 5: Thank you for your advice. We added some references of the cognitive tests in the article and further described these tests. The details are as follows:

“As for domain-specific cognitive tests, words in DST backwards and DST forwards are summed to obtain DST total score. RAVLT in our study includes 3 sub-tests: RAVLT total learning, RAVLT long-delayed recall, and RAVLT recognition. ROCF has 4 sub-tests, including ROCF copy, ROCF immediate recall, ROCF long-delayed recall, and ROCF recognition [29]. The Victoria version Stroop tests consists of 3 tests, Dot (D) test, Word (W) test, and Color (C) test. Time to complete each test is scored [30]. The obtained score of TMT A and TMT B represent the amount of time required to complete the task [31]. The score of SDMT is the number of correctly matched digits and symbols in a given time (90 seconds) [32]. VFT is determined by the number of animal names in 1 minute. BNT Beijing version includes 30 pictures, and the number of correctly named pictures are scored. CDT requires subjects to draw a clock pointing to 11:10 on a white sheet of paper [33].”

Point 6: (Results) How many patients were excluded from participants of CANDOR?

Response 6: CANDOR is an onging study which has 3 cohorts, cognitive normal cohort, Alzheimer's disease cohort, and vascular cognitive impairment cohort (https://clinicaltrials.gov/ct2/show/record/NCT04320368?view=record). The patients included in this study are all from the vascular cognitive impairment cohort. Based on the eligibility criteria, data of a total of 414 patients were obtained. After excluding individuals without the information of monthly income, and those with missing 3D-T1 MRI data and incomplete neuropsychological assessment, 294 patients were finally enrolled in this study.

Point 7: (Results, line 149) “Nighty-four”

Response 7: Thank you for your pointing this out. The “Nighty-four” has been corrected into “ninety-four”

Point 8: (Discussion) As the authors wrote in the limitation, causal association is uncertain in this study. Therefore, “effect” in the title should be replaced with “association”.

Response 8: Thanks for your advice. Based on your suggestion, we have replaced the ”effect” with “association” in the title.

Thank you very much for your attention and time. Look forward to hearing from you.

Yours sincerely,

1st Feb. 2023